# The Influence of Indisulam on Human Immune Effector Cells: Is a Combination with Immunotherapy Feasible?

**DOI:** 10.3390/pharmaceutics17030368

**Published:** 2025-03-14

**Authors:** Lisa Arnet, Lisabeth Emilius, Annett Hamann, Maria Carmo-Fonseca, Carola Berking, Jan Dörrie, Niels Schaft

**Affiliations:** 1Department of Dermatology, Universitätsklinikum Erlangen, Friedrich-Alexander-Universität ErlangenNürnberg, 91054 Erlangen, Germany; lisa.arnet@fau.de (L.A.); lisabeth.emilius@meduniwien.ac.at (L.E.); annett.hamann@uk-erlangen.de (A.H.); carola.berking@uk-erlangen.de (C.B.); jan.doerrie@uk-erlangen.de (J.D.); 2Comprehensive Cancer Center Erlangen European Metropolitan Area of Nuremberg (CCC ER-EMN), 91054 Erlangen, Germany; 3Deutsches Zentrum Immuntherapie (DZI), 91054 Erlangen, Germany; 4Bavarian Cancer Research Center (BZKF), 91054 Erlangen, Germany; 5Instituto de Medicina Molecular João Lobo Antunes, Faculdade de Medicina da Universidade de Lisboa, 1649-028 Lisbon, Portugal; carmo.fonseca@medicina.ulisboa.pt

**Keywords:** indisulam (E7070), neoantigens, immunotherapy, T cells, dendritic cells

## Abstract

**Background:** As a modulator of pre-mRNA splicing, the anti-cancer agent indisulam can induce aberrantly spliced neoantigens, enabling immunologic anti-tumor activity. Consequently, combining indisulam with immunotherapy is expected to be a promising novel approach in cancer therapy. However, a prerequisite for such a combination is that immune effector cells remain functional and unharmed by the chemical. **Methods:** To ensure the immunocompetence of human immune effector cells is maintained, we investigated the influence of indisulam on ex vivo-isolated T cells and monocyte-derived dendritic cells (moDCs) from healthy donors. We used indisulam concentrations from 0.625 µM to 160 µM and examined the impact on the following: (i) the activation of CD4^+^ and CD8^+^ T cells by CD3-crosslinking and via a high-affinity TCR, (ii) the cytotoxicity of CD8^+^ T cells, (iii) the maturation process of moDCs, and (iv) antigen-specific CD8^+^ T cell priming. **Results:** We observed dose-dependent inhibitory effects of indisulam, and substantial inhibition occurred at concentrations around 10 µM, but the various functions of the immune system exhibited different sensitivities. The weaker activation of T cells via CD3-crosslinking was more sensitive than the stronger activation via the high-affinity TCR. T cells remained capable of killing tumor cells after treatment with indisulam up to 40 µM, but T cell cytotoxicity was impaired at 160 µM indisulam. While moDC maturation was also rather resistant, T cell priming was almost completely abolished at a concentration of 10 µM. **Conclusions:** These effects should be considered in possible future combinations of immunotherapy with the mRNA splicing inhibitor indisulam.

## 1. Introduction

Cancer immunotherapy, which aims to induce the patient’s immune system to recognize and kill cancer cells, has shown great potential in recent years and has been hailed as a therapeutic breakthrough. However, in spite of excellent therapeutic achievements, the last few years have also revealed that not all cancers and patients can benefit from the available immunotherapies [1]. Therefore, the enhancing and broadening of the effectiveness of immunotherapy is a subject of current research [2].

Cancer is mainly driven by somatic DNA mutations [3], which can cause abnormal regulation of alternative splicing and can lead to the presentation of tumor-specific molecules on the surface of cancer cells, namely neoantigens [4]. The immune system can identify neoantigens as foreign proteins and trigger a tumor-targeted cellular immune response. Boosting the immune system to recognize such cancerous changes and attack malignant cells is one of the key principles of immunotherapeutic strategies [5]. Neoantigens play a central role in this context, allowing selective targeting and killing of cancer cells. These approaches are known as neoantigen-based therapies and include, in particular, adoptive cell therapy using tumor-infiltrating lymphocytes, T cell receptor-engineered T cells, or chimeric antigen receptor T cells, antibody-based therapy, and therapeutic vaccines including peptide, nucleic acid, or dendritic cell vaccines [6]. The effectiveness of immunotherapy correlates with a high mutational burden, implying the formation of neoantigens [7]. Neoantigens can arise intrinsically by aberrant splicing due to mutations in the splicing machinery [8], but can also be induced pharmacologically by splicing inhibitors, e.g., indisulam [4,9,10]. In this way, the expression of neoantigens can be therapeutically prompted, which gives hope in the fight against cancer.

Indisulam (or E7070) is an aryl sulfonamide and belongs to the group of splicing inhibitor sulfonamides (SPLAM). It causes ubiquitination and proteasome-mediated degradation of the mRNA splicing factor RBM39 (RNA-binding motif protein 39) [11]. A study conducted by Lu et al. showed that administering low doses of indisulam reduced RBM39 levels in cancer cells without suppressing their growth [10]. However, when these cells were engrafted into immunocompetent mice, tumor growth was inhibited. Additional experiments, both in vitro and in vivo, indicated that treatment with indisulam induced the production of splicing-derived neoantigens that trigger an endogenous T cell response. These findings imply that indisulam can modulate splicing in a manner that generates immunogenic neoantigens.

Combining immunotherapy with indisulam-induced neoantigen formation may offer a promising new approach for the development of more effective immunotherapeutic strategies against cancer, especially in cases that show limited sensitivity to conventional therapies. Since indisulam not only affects the tumor cells, but potentially also all other cells in the human body, and has both splice inhibition-related effects and, at higher concentrations, unspecific toxicity, while immunotherapies rely on the patient’s own immune system, it must be ensured that the immune cells remain functional in the presence of indisulam. The before-mentioned study from Lu et al. initiated research on the influence of indisulam on mouse T cells, finding that indisulam was largely well tolerated [10]. However, there is hardly any data on the impact of this splicing inhibitor on human immune effector cells. Here, we address this lack using ex vivo-isolated CD4^+^ and CD8^+^ T cells and moDCs from healthy donors to evaluate the dose-dependent effects of indisulam on T cell activation, T cell cytotoxicity, moDC maturation, and antigen-specific T cell priming.

This study aims to determine the level of indisulam tolerated by human immune effector cells and to monitor which T cell functions are compromised at which inhibitor concentrations to provide insights into the potential feasibility of future combinations of immunotherapy with the splicing modulator indisulam.

## 2. Materials and Methods

### 2.1. Indisulam (E7070)

The splicing inhibitor indisulam was obtained from Sigma Aldrich (St. Louis, MO, USA) and dissolved in dimethyl sulfoxide (DMSO, Sigma Aldrich, St. Louis, MO, USA) according to the manufacturer’s instructions. In the course of our experiments, we switched from a 10 mM indisulam stock to a 50 mM indisulam stock to reduce the volume of indisulam added to the cells and therefore minimize a possible effect of the solvent (DMSO), to ensure that the observed effects were not DMSO-dependent but indisulam-specific.

### 2.2. Media and Buffers

The R10 medium consisted of Roswell Park Memorial Institute (RPMI) 1640 (Lonza, Verviers, Belgium) supplemented with 10% heat-inactivated fetal calf serum (Sigma Aldrich, St. Louis, MO, USA), 1% L-glutamine (200 mM, Lonza, Verviers, Belgium), 0.2% HEPES buffer (1 M, Lonza, Verviers, Belgium), 0.04% gentamycin (500 mg/mL, Life Technologies, Karlsbad, CA, USA), and 0.04% β-mercaptoethanol (50 mM, Roth, Karlsruhe, Germany). This medium was used for the cultivation of T cells.

DC medium consisted of RPMI 1640 supplemented with 1% heat-inactivated human serum (Sigma Aldrich, St. Louis, MO, USA), 1% L-glutamine (200 mM), and 0.04% gentamycin (50 mg/mL). This medium was used for the generation procedure of moDCs.

MLPC medium consisted of RPMI 1640 supplemented with 10% heat-inactivated human serum, 1% L-glutamine (200 mM), 1% sodium pyruvate (200 mM, Lonza, Verviers, Belgium), 1% HEPES buffer (1 M), 1% non-essential amino acids (200 mM, Lonza, Verviers, Belgium), and 0.04% gentamycin (50 mg/mL). This medium was used for the co-culture of T cells with DCs during the one-week priming protocol.

MACS buffer consisted of Dulbecco’s phosphate-buffered saline (DPBS, Lonza, Verviers, Belgium) supplemented with 2.5% human serum albumin (20%, Bio & SELL GmbH, Feucht/Nuremberg, Germany).

FACS buffer consisted of DPBS supplemented with 1% heat-inactivated fetal calf serum and 0.02% sodium azide (Lonza, Verviers, Belgium). This medium was used for diluting fluorescently labeled antibodies and flow cytometric measurements.

ELI medium consisted of RPMI 1640 supplemented with 5% heat-inactivated human serum, 1% L-glutamine, 1% HEPES buffer (1 M), and 0.04% gentamycin (50 mg/mL). This medium was used for major histocompatibility complex tetramer staining.

### 2.3. Cells

#### 2.3.1. Human Immune Effector Cells

The human immune effector cells used in all our experiments were isolated and generated from fresh whole blood of healthy donors after written informed consent and approval by the institutional review board were obtained. Peripheral blood mononuclear cells (PBMCs) were purified by density gradient centrifugation using Lymphoprep reagent (Cosmo Bio, Carlsbad, CA, USA) and then separated into a non-adherent fraction (NAF) and monocytes by plastic adherence as described previously in detail [12]. All cells were incubated at 37 °C with 5% CO_2._

##### Ex Vivo-Isolated CD4^+^ or CD8^+^ T Cells

CD4^+^ or CD8^+^ T cells were isolated from NAF by using anti-CD4 or anti-CD8 magnetic-activated cell sorting (MACS) beads (Miltenyi Biotec, Bergisch Gladbach, Germany) following the manufacturer’s instructions. During this procedure, cells were kept in MACS buffer. After isolation, T cells were used in further experimental setups as described below in the corresponding sections.

##### Monocyte-Derived Dendritic Cells (moDCs)

Monocytes were differentiated into immature DCs following an incubation protocol over the course of 6 days in DC medium, as described elsewhere [12]. On days 1, 3, and 5, fresh DC medium, plus 800 U/µL granulocyte-macrophage colony-stimulating factor (GM-CSF, PeproTech, Cranbury, NJ, USA) and 275 U/µL interleukin (IL)-4 (PeproTech, Cranbury, NJ, USA) were added. On day 5 (approximately 24 h before maturation), DCs were treated with indisulam or DMSO as indicated (see Section 2.7), or were left untreated as a control. On day 6, immature DCs were matured with a cytokine cocktail containing 200 U/µL IL-1β (CellGenix GmbH, Freiburg, Germany), 1 µg/mL PGE_2_ (Santa Cruz Biotechnology, Dallas, TX, USA), 1000 U/µL IL-6 (CellGenix GmbH, Freiburg, Germany), and 10 ng/mL TNFα (PeproTech, Cranbury, NJ, USA). After approximately 24 h, mature DCs were harvested and either analyzed directly or used in further experiments (see Section 2.7, Section 2.8 and Section 2.12).

#### 2.3.2. T2.A1 Cells

T2.A1 cells (TAP-deficient TxB cell hybrid) were used as peptide-loaded target cells in antigen-specific T cell activation experiments as described below (see Section 2.6).

### 2.4. Treatment with Indisulam

Unless otherwise specified, cells were treated in vitro either with indisulam, DMSO solvent control, or were left untreated as a reference value (no indisulam, no DMSO). For this purpose, an appropriate quantity of indisulam, DMSO solution, or R10 medium was added to the respective conditions. After approximately 24 h or 48 h, cells were utilized for subsequent experiments, including non-specific T cell activation, antigen-specific T cell activation, T cell cytotoxicity, maturation of moDCs, and antigen-specific T cell priming.

### 2.5. Treatment with Indisulam and Non-Specific Activation of T Cells

After isolation, CD4^+^ and CD8^+^ T cells were divided into eight conditions in R10 medium plus 10 ng/mL IL-7 (PeproTech, Cranbury, NJ, USA). To assess the influence of indisulam on non-specific T cell activation, T cells were either treated with indisulam at different concentrations over a wide range (0.625 µM, 2.5 µM indisulam, 10 µM indisulam, 40 µM indisulam, 160 µM indisulam), treated with DMSO solvent controls (DMSO low, equivalent to the quantity of DMSO contained in 40 µM indisulam and DMSO high, equivalent to the quantity of DMSO contained in 160 µM indisulam), or left untreated as a reference value (no indisulam, no DMSO). For all donors, the 50 mM indisulam stock was used. After treatment of approximately 24 h or 48 h, T cells were split into two wells. One well of each condition was left unactivated as a control, and the other well was activated by applying 0.1 µg/mL OKT-3 (inducing CD3-crosslinking, Miltenyi Biotec, Bergisch-Gladbach, Germany) and 1000 U/mL IL-2 (Novartis, Basel, Switzerland). After approximately 20–24 h of activation, supernatants were sampled, and cytokine secretion was measured by BD™ Cytometric Bead Array (CBA) human Th1/Th2 cytokine kit II (BD Biosciences, San José, CA, USA) (see Section 2.9). The cells were harvested and stained for the activation markers CD25 and CD69 (see Section 2.8). In addition, a live/dead staining was performed with a 7-AAD-PerCP antibody (Thermo Fisher Scientific Inc., Waltham, MA, USA). Finally, the cells were analyzed by flow cytometry.

### 2.6. Treatment with Indisulam and Antigen-Specific Activation of T Cells

In general, our antigen-specific activation was performed with CD4^+^ or CD8^+^ T cells equipped with a gp100-specific T cell receptor (TCR) and activated by co-incubation with target cells loaded with the corresponding peptide as described previously [13]. This procedure involved the following steps: After isolation, CD4^+^ and CD8^+^ T cells were first cultured in R10 medium plus 10 ng/mL IL-7 and then rested overnight to allow degeneration of MACS beads. The next day, T cells were harvested and prepared for antigen-specific activation. Enabling antigen-specific T cell activation, T cells were electroporated with two RNAs encoding the α- and β-chain of the gp100-specific TCR as described previously [14,15]. Afterwards, to assess the influence of indisulam on antigen-specific T cell activation, electroporated T cells were each separated into eight conditions and then treated either with indisulam at different concentrations (0.625 µM indisulam, 2.5 µM indisulam, 10 µM indisulam, 40 µM indisulam, 160 µM indisulam), treated with the DMSO solvent controls (DMSO low (equivalent to the quantity of DMSO contained in 40 µM indisulam) and DMSO high (equivalent to the quantity of DMSO contained in 160 µM indisulam)), or left untreated as a reference value (no indisulam, no DMSO). For all donors, the 50 mM indisulam stock was used. After treatment of approximately 24 h or 48 h, the target cells, in our case T2.A1 cells, were either loaded with gp100-peptides (10 µg/mL) for 1 h at 37 °C or left native (unloaded) as unactivated controls. Subsequently, electroporated T cells and peptide-loaded or native T2.A1 cells were co-cultured at a 1:1 ratio in R10 medium. During the co-culture, the same conditions (treatment with indisulam, treatment with DMSO solution, or no treatment) as before were kept. After 20–24 h of activation in the presence or absence of indisulam or DMSO, supernatants were sampled, and cytokine secretion was measured by BD™ Cytometric Bead Array (CBA) human Th1/Th2 cytokine kit II (see Section 2.9). The cells were harvested and stained for the activation markers CD25 and CD69 (see Section 2.8). In addition, a live/dead staining was performed with a 7-AAD-PerCP antibody (Thermo Fisher Scientific Inc., Waltham, MA, USA). Finally, the cells were analyzed by flow cytometry.

### 2.7. Treatment with Indisulam and moDC Maturation Process

When separating the PBMCs into non-adherent fraction (NAF) and monocytes, the PBMCs were first divided into the different treatment conditions and then incubated for 1 h at 37 °C with 5% CO_2_. After collecting the NAF, the adherent monocytes were differentiated into immature DCs. The maturation stimulant (cytokine cocktail) was added on day 6. To see if the presence of indisulam affects their maturation process, immature moDCs were either treated with indisulam at different concentrations (0.625 µM indisulam, 2.5 µM indisulam, 10 µM indisulam, 40 µM indisulam) or treated with DMSO solvent control (DMSO equivalent to the quantity of DMSO contained in 40 µM indisulam) on day 5 (24 h before adding the maturation stimulant). For 3 out of 4 donors, moDCs were treated with the 10 mM indisulam stock. For one donor, we used the 50 mM indisulam stock and we additionally left a moDC sample untreated as a reference value (no indisulam, no DMSO). For surface marker expression analysis, after approximately 24 h of maturation by cytokine cocktail in the presence or absence of indisulam or DMSO, the moDCs were harvested and stained for the maturation markers CD25, CD83, CD86 and PD-L1 (see Section 2.8). In addition, a live/dead staining was performed with a 7-AAD-PerCP antibody. Finally, the moDCs were analyzed by flow cytometry.

### 2.8. Surface Marker Expression Analysis

In the case of T cell activation, extracellular surface marker staining was performed with anti-CD25-FITC (BD Biosciences, San José, CA, USA) and anti-CD69-PE (BD Biosciences, San José, CA, USA). In the case of moDC maturation, extracellular surface marker staining was performed with anti-CD25-FITC, anti-CD83-PE (Miltenyi Biotec, Bergisch-Gladbach, Germany), anti-CD86-FITC (BD Biosciences, San José, CA, USA), and anti-PD-L1-PE (eBioscience, Thermo Fisher Scientific, San Diego, CA, USA). The antibodies were diluted in FACS buffer and left on the cells for 30 min at 4 °C in the dark. All samples were measured by FACSCanto II Flow Cytometer (BD Biosciences, San José, CA, USA) and evaluated by FlowJo, version 10.9 (BD Biosciences, San José, CA, USA). The surface markers expression is indicated as specific MFI (mean fluorescence intensity), which was calculated by subtracting the background MFI of the unstained controls. In the case of T cell activation, the expression level of the respective activated, untreated sample was considered as 100% activation, and all other values were normalized to this condition; i.e., the figures display the relative expression of activation markers (specific MFI) in relation to their reference value (no indisulam, no DMSO). Additionally, the data showing the specific MFI and the percentage positive population are provided in the Appendix A.

### 2.9. Cytokine Analysis

To quantify the cytokine secretion, the supernatants of the samples were collected 20–24 h after T cell activation. The cytokine secretions of T cells were quantified by using the BD™ Cytometric Bead Array (CBA) human Th1/Th2 cytokine kit II as instructed by the manufacturer. The samples were measured by FACSCanto II and evaluated by FCS Express software, version 5 (DeNovo Software, Pasadena, CA, USA). Afterwards, the cytokine concentrations were calculated using the calibration line based on the standard dilution series. The secretion level of the respective activated, untreated sample was considered as 100% activation and all other levels were normalized to this condition; i.e., the figures display the relative cytokine concentration in relation to their reference value (no indisulam, no DMSO). Additionally, the data showing the absolute cytokine concentration are provided in the Appendix A.

### 2.10. Treatment with Indisulam and T Cell Cytotoxicity

After magnetic cell isolation, CD8^+^ T cells were rested overnight in R10 medium, plus 10 ng/mL IL-7. The following day, half of the T cells were transfected with a gp100-specific TCR, enabling antigen-specific T cell cytotoxicity, and the other half was left unequipped, serving as mock controls. Concretely, T cells were either electroporated with two RNAs encoding the α- and β-chain of the gp100-specific TCR (T cells^gp100^) or electroporated without RNA (T cells^mock^). After 2–4 h of rest, T cells^gp100^ and T cells^mock^ were each divided into eight conditions and then treated for approximately 24 h either with indisulam at different concentrations (0.625 µM indisulam, 2.5 µM indisulam, 10 µM indisulam, 40 µM indisulam, 160 µM indisulam), or with DMSO solvent controls (DMSO low (equivalent to the quantity of DMSO contained in 40 µM indisulam) and DMSO high (equivalent to the quantity of DMSO contained in 160 µM indisulam)), or left untreated as reference values (no indisulam, no DMSO). For all donors, the 50 mM indisulam stock was used. The next day, the cytotoxicity assay was set up. Tumor cells (A375M) loaded with gp100-peptides (10 µg/mL) for 1 h at 37 °C were used as target cells, while unloaded tumor cells (A375M) served as control cells. The target cells were labeled with 3.0 µM Carboxyfluorescein succinimidyl ester (CFSE, Invitrogen, ThermoFisher Scientific, Eugene, OR, USA) (diluted in PBS) and the control cells with 0.25 µM CFSE, and both were mixed at a 1:1 ratio. Afterwards, tumor cells were either left alone as reference values or were co-incubated with T cells^gp100^ or with T cells^mock^ at a 1:10 ratio in R10 medium. During the co-culture, the same conditions (treatment with indisulam, treatment with DMSO solution, or no treatment) were kept as before. After 18–20 h of co-incubation, the cells were harvested and analyzed by flow cytometry (see Section 2.11).

### 2.11. Cytotoxicity Analysis

Cells were harvested and stained with the live/dead marker Zombie NIR (BioLegend, San Diego, CA, USA), which was diluted in FACS buffer and left on the cells for 20 min at 4 °C in the dark. All samples were measured by FACSCanto II and evaluated by FlowJo, version 10.9 (BD Biosciences, San José, CA, USA). Data analysis was performed by determining the percentage of living target tumor cells (Zombie NIR-negative and CFSE^high^) and living control tumor cells (Zombie NIR-negative and CFSE^low^). The figures show the percentage of tumor lysis, which was calculated using the following formula:$$A=\frac{\%\;living\;target\;tumor\;cells}{\%\;living\;control\;tumor\;cells}$$$$\%\;of\;lysis=\frac{A\;\%\left[tumor\;cells\;alone\right]-A\;\%[tumor\;cell+{T\;cells}^{gp100\;or\;mock}]}{A\;\%[tumor\;cells\;alone]}\times 100\%$$

### 2.12. Treatment with Indisulam and Antigen-Specific T Cell Priming

The effect of indisulam on antigen-specific CD8^+^ T cell priming by peptide-loaded moDCs was investigated. In preparation for the priming approach, monocytes were isolated from the blood of healthy donors and then differentiated into moDCs following a one-week protocol as described above. One day before maturation with a cytokine cocktail, moDCs were treated with indisulam at different concentrations (0.625 µM indisulam, 2.5 µM indisulam, 10 µM indisulam, 40 µM indisulam) or treated with DMSO solvent control (DMSO equivalent to the quantity of DMSO contained in 40 µM indisulam). After approximately 24 h of maturation, moDCs from each treatment condition were harvested and loaded with a MelanA wildtype peptide (EAAGIGILTV, HLA-A*02:01), which is frequently used and has proven to be strongly immunogenic [16]. For this, moDCs were incubated with 10 µL/mL MelanA peptide for 1 h at 37° C. Unloaded moDCs served as negative controls to verify the method worked and to calculate the average background later in the analysis. On the same day, blood was drawn again from the same donor. This time, NAF containing CD8^+^ T cells was collected and utilized for the priming approach. The priming approach was performed with moDCs and NAF at a ratio of 1:10–1:20 in MLPC medium plus 10 ng/mL IL-7. First, the NAF was divided into the different treatment conditions, then an appropriate amount of indisulam or DMSO was applied, and finally the moDCs were added. The following conditions were tested: 0.625 µM indisulam, 2.5 µM indisulam, 10 µM indisulam, 40 µM indisulam, and DMSO solvent control (DMSO equivalent to the quantity of DMSO contained in 40 µM indisulam). Two donors were treated with the 10 mM indisulam stock. For four donors, we used the 50 mM indisulam stock and an additional untreated sample was tested to evaluate whether the observed effects were not DMSO-dependent, but indisulam-specific. The stimulation was incubated at 37 °C for 7 days. On day 2 and 4, 1000 U/mL IL-2 (Novartis, Basel, Switzerland) and 10 ng/mL IL-7 (Peprotech, Cranbury, NJ, USA) were added. On day 7, the cells were harvested and used for tetramer staining (see Section 2.13).

### 2.13. Major Histocompatibility Complex Tetramer Staining

After being primed by moDCs for one week, the CD8^+^ T cells were analyzed for antigen specificity. For this, the cells from autologous stimulation were stained with fluorescently labeled MelanA/MHC tetramer (Biozol, Hamburg, Germany) or dextramer (Immudex, Virum, Denmark). The MHC multimer was diluted in ELI medium. For staining, the tetramer was left on the cells for 30 min at room temperature in the dark. Subsequently, the cells were stained with the fluorescently labeled antibodies anti-CD8-PE-Cy7 (BD Biosciences, San José, CA, USA), anti-CD4-V500 (BD Biosciences, San José, CA, USA), anti-CCR7-FITC (BioLegend, San Diego, CA, USA), and anti-CD45RA-Brilliant Violet 421 (BioLegend, San Diego, CA, USA). All antibodies were diluted in FACS buffer. For staining, the antibodies were left on the cells for 30 min at 4 °C in the dark. Just before measuring by FACSCanto II Flow Cytometer (BD Biosciences, San José, CA, USA), the live/dead antibody 7-AAD-PerCP was added to each sample. The percentage of antigen-specific CD8^+^ T cells and their phenotype were analyzed by FlowJo, version 10.9 (BD Biosciences, San José, CA, USA). The figures show the percentage of the respective population.

### 2.14. Statistical Analysis and Data Visualization

Statistical analysis and data visualization were performed using GraphPad Prism version 10.2.0 (GraphPad Software, La Jolla, CA, USA). The influence of indisulam was evaluated by comparing the different indisulam concentrations with the respective DMSO solvent controls. Thus, the conditions with 0.625 µM, 2.5 µM, 10 µM, and 40 µM indisulam were tested compared to DMSO low (treatment with the quantity of DMSO that was contained in the respective condition with 40 µM indisulam). The condition with 160 µM indisulam was tested compared to DMSO high (treatment with the quantity of DMSO that was contained in the respective condition with 160 µM indisulam). Statistical significance and *p*-values were calculated by a paired Student’s *t*-test, assuming normal Gaussian distribution and equal standard deviation, based on our experience in similar experiments. For *p*-values above 0.05, differences between groups were considered not significant (ns). If *p* was smaller than 0.05, differences between groups were considered significant (* *p* ≤ 0.05, ** *p* ≤ 0.01, *** *p* ≤ 0.001, **** *p* ≤ 0.0001). Please note that not all formal requirements for the paired Student’s *t*-test were achieved in our setting: Gaussian distribution could not be tested due to our limited sample size. Nevertheless, it can be assumed that the *t*-test is quite robust [17,18]. Details for the statistical methods used in the individual experiments are given in the figure legends.

## 3. Results

### 3.1. Influence of Indisulam on CD4^+^ and CD8^+^ T Cell Activation

T cells are key components of the adaptive immune system that mediate cell-based immune responses to fight against intracellular pathogens and eliminate malignant cancer cells [17,18]. To ensure that the T cells were still functional in the presence of indisulam, we first examined if indisulam impedes the activation of CD4^+^ and CD8^+^ T cells. For this purpose, ex vivo-isolated CD4^+^ and CD8^+^ T cells were either left untreated (served as reference value), or treated with DMSO (served as solvent control), or treated with indisulam at the following different concentrations: 0.625 µM, 2.5 µM, 10 µM, 40 µM, or 160 µM. CD4^+^ T cells were treated for 24 h, while CD8^+^ T cells were treated for 24 h and 48 h. Since indisulam was dissolved in DMSO according to the manufacturer’s instructions, and the quantity of DMSO increased accordingly with higher indisulam concentrations, we performed two DMSO solvent controls at different concentrations. DMSO low (equivalent to the quantity of DMSO contained in 40 µM indisulam) served as the solvent control for the conditions with 0.625 µM–40 µM indisulam. DMSO high (equivalent to the quantity of DMSO contained in 160 µM indisulam) served as the solvent control for the highest indisulam concentration of 160 µM. After treatment, the T cells were stimulated either with (i) non-specific activation or (ii) antigen-specific activation. The non-specific activation of CD4^+^ and CD8^+^ T cells was induced by CD3-crosslinking with OKT-3 (anti-CD3 antibody). The antigen-specific activation was performed with CD4^+^ or CD8^+^ T cells transfected with a gp100-specific T cell receptor (TCR) and co-incubated with gp100-loaded target cells, namely, T2.A1 cells. Non-activated cells from each sample served as negative controls to ensure that activation was induced. Following 24 h of activation, the supernatants were collected and tested for secreted cytokines, measured by Cytometric Bead Array (CBA). The T cells were harvested, and living cells were analyzed for the surface expression of the activation markers CD25 and CD69 by flow cytometry. In this context, it must be noted that the non-specific T cell activation by CD3-crosslinking represented a weaker stimulus than the antigen-specific activation via the gp100-specific high-affinity TCR, as exemplarily shown by CD69 absolute expression levels in Figure 1.

#### 3.1.1. The Expression of Activation Markers on T Cells Is Inhibited by High Indisulam Concentrations

The assessment of non-specific activation of T cells showed a correlation between increased doses of indisulam and decreased upregulation of CD25 or CD69 surface expression (Figure 2, Appendix A). Regarding CD4^+^ T cells, the expression of CD25 and CD69 remained stable with concentrations of up to 2.5 µM and 10 µM indisulam, respectively. However, the upregulation of both activation markers was significantly reduced at higher indisulam concentrations (Figure 2a, Appendix A). Regarding CD8^+^ T cells, the expression of CD25 and CD69 had already decreased at a dose of 0.625 µM indisulam. In general, the effects were stronger with the 48 h indisulam treatment (Figure 2b, Appendix A).

The evaluation of gp100-specific activation also revealed dose-dependent effects of indisulam, but these were less pronounced (Figure 3, and Appendix A). In addition, a stronger donor variance was observed. Considering the mean values for the activation marker expression on CD4^+^ and CD8^+^ T cells after 24 h indisulam treatment, the gp100-specific activation seemed to decrease mainly at the highest indisulam concentrations (Figure 3, Appendix A). However, regarding the 48 h pretreated-CD8^+^ T cells, the expression of CD25 gradually decreased with increasing indisulam concentration (Figure 3b, Appendix A).

In summary, for non-specific and gp100-specific T cell activation, little to no differences in the upregulation of CD25 and CD69 were observed in the DMSO solvent controls compared to the untreated samples (displayed as relative surface expression of about 1). This suggested that the detected effects were caused by indisulam treatment (Figure 2 and Figure 3). Overall, the surface marker experiments showed that indisulam had a dose-dependent negative effect on T cell activation and that a weaker activation method was more prone to inhibition, while a stronger activation via a high-affinity TCR was rather resistant. In addition, CD8^+^ T cells seemed slightly more sensitive than CD4^+^ T cells. With increasing durations of indisulam treatment, the effects were more prominent.

#### 3.1.2. Indisulam Has Dose-Dependent and Time-Dependent Effects on Cytokine Secretion by Activated T Cells

In addition to the measurement of surface activation markers, we sampled the supernatants for T cell cytokines after the activation of the T cells. In line with our data from evaluating CD25 and CD69 surface expression levels, tests with indisulam indicated dose-dependent and time-dependent effects on cytokine secretion by OKT-3-activated T cells (Figure 4, Appendix A). Please note that IL-2 could not be analyzed, since recombinant IL-2 is added as part of the activation protocol. In the case of 24 h indisulam treatment, regarding CD4^+^ T cells, the reduced secretion of IL-10 was statistically significant at indisulam doses of at least 10 µM, and that of IL-6 and IFNγ at 40 µM. TNFα secretion was also impaired at these indisulam levels, but reached no formal statistical significance in comparison to the respective controls (Figure 4a, Appendix A). Regarding CD8^+^ T cells, IFNγ release was already significantly decreased at the lowest concentration of indisulam (0.625 µM) and above. Again, the changes in TNFα secretion did not reach statistical significance. In the case of 48 h indisulam treatment, CD8^+^ T cells almost completely ceased their IFNγ and TNFα secretion (Figure 4b, Appendix A). IL-10 and IL-6 were only produced in marginal concentrations by the CD8^+^ T cells.

In contrast to the weaker activation stimulus by OKT-3 activation, the effects on the gp100-specific T cell cytokine secretion were less definite (Figure 5, Appendix A), similar to what we had observed for the expression of activation markers, and for most cytokines only the highest concentrations showed a reduction in cytokine secretion after 24 h of indisulam pretreatment. In the case of CD4^+^ T cells, this was only statistically significant for decreased IL-10 secretion, but the same tendency was observed for IFNγ, TNFα, IL-2, and IL-6 (Figure 5a, Appendix A). In the case of CD8^+^ T cells, TNFα and IL-2 secretion was also markedly impaired after pretreatment with 160 µM indisulam, as shown by the mean values. IFNγ release was already significantly reduced at an indisulam dose of 10 µM and above. In line with the results from the non-specific T cell activation, IFNγ and TNFα secretion by CD8^+^ T cells decreased drastically after 48 h indisulam treatment. However, IL-2 release remained relatively stable even after the 48 h pretreatment, but declined again at the highest indisulam dose (Figure 5b, Appendix A). IL-10 and IL-6 were only produced in marginal concentrations by the CD8^+^ T cells.

Looking at the DMSO-treated T cells, it was notable that the cytokine secretions were partly lower than in the untreated samples. This indicated that DMSO also had an impact on cytokine secretion, and that the observed effects are not only due to the influence of indisulam. This effect was especially high for TNFα. It should also be mentioned that the variability in cytokine release between the different donors was relatively high (Figure 4 and Figure 5).

Overall, the results from the cytokine measurements followed those from the activation markers. Accordingly, cytokine release was more prone to indisulam-mediated inhibition when the activation signal was weaker (via unspecific CD3-crosslinking), and less affected, but also of higher inter-donor-variance upon a stronger activation (via a high-affinity TCR). Moreover, a longer treatment with indisulam resulted in a much stronger impairment of cytokine secretion for both non-specific and antigen-specific CD8^+^ T cell activation, as indicated by the results obtained after 48 h of indisulam treatment.

### 3.2. Influence of Indisulam on T Cell Cytotoxicity

The most important function of CD8^+^ T cells is their antigen-specific cytotoxicity. Hence, we investigated the capacity of T cells to kill cancer cells after pretreatment with indisulam. Pursuing the idea of combining immunotherapy with neoantigen generation by indisulam treatment, it must be ensured that the T cells remain capable of recognizing and killing tumor cells.

#### The Capacity of T Cells to Kill Tumor Cells Proves to Be Robust

To determine the extent of tumor lysis due to T cells, we performed a cytotoxicity assay with ex vivo-isolated CD8^+^ T cells, which were either transfected with a gp100-specific TCR enabling antigen-specific T cell cytotoxicity (T cells^gp100^) or left unequipped serving as mock controls (T cells^mock^). To explore the influence of indisulam, T cells were either left untreated, treated with DMSO or treated with indisulam at different concentrations: 0.625 µM, 2.5 µM, 10 µM, 40 µM, or 160 µM. As described above, we again included two DMSO solvent controls. DMSO low (equivalent to the quantity of DMSO contained in 40 µM indisulam) served as solvent control for the conditions with 0.625 µM–40 µM indisulam. DMSO high (equivalent to the quantity of DMSO contained in 160 µM indisulam) served as solvent control for the highest indisulam concentration of 160 µM. After 24 h pretreatment with indisulam or DMSO, T cells^gp100^ and Tcells^mock^ were co-cultured (at a 1:10 ratio) with a tumor cell mix consisting in equal parts of antigen-positive (gp100-positive) target tumor cells and antigen-negative (gp100-negative) control tumor cells. After co-incubating overnight in the presence or absence of indisulam or DMSO, the cells were harvested, stained with a live/dead marker and analyzed by flow cytometry. Finally, the percentage of tumor lysis was calculated. Indeed, the lytic capacity of the CD8^+^ T cells appeared relatively robust to the influence of indisulam up to a concentration of 2.5 µM, with only marginal effects at 10 µM, and even following treatment with 40 µM indisulam, the antigen-specific T cells (Tcells^gp100^) were still able to kill the antigen-positive tumor cells. Only at the highest concentration of 160 µM indisulam was tumor killing diminished (Figure 6). These results indicate that the eponymous function of cytotoxic T lymphocytes is only compromised at very high concentrations of the inhibitor.

### 3.3. Other Effects of Indisulam on T Cells

#### Increasing Concentrations of Indisulam Cause a Decrease in Cell Size Without a Relevant Increase in Cell Death

To investigate whether indisulam had general toxic effects on CD4^+^ and CD8^+^ T cells, we determined the number of living cells after 48 h or 72 h treatments with DMSO or indisulam at different concentrations. This evaluation was performed for both unactivated T cells and T cells activated with OKT-3 for the last 24 h of indisulam treatment. The vitality was measured by staining the cells with the live/dead marker 7-AAD and subsequent flow cytometric analysis. The 7-AAD-negative cells were identified as living cells. Regarding the mean values for the percentage of living cells among all cells, T cell vitality was not impaired by indisulam treatment up to a concentration of 40 µM, but the highest indisulam concentration of 160 µM caused an increase in cell death (Figure 7). This effect was similar for both CD4^+^ (Figure 7a) and CD8^+^ (Figure 7b) T cells, and survival was decreased more considerably after 72 h, but, again, only in the 160 µM condition. This indicated that only at the highest concentration of 160 µM could the various effects described above be attributed to the general toxicity of indisulam.

Remarkably, in our flow cytometric analyses, we observed that indisulam led to a shrinkage of T cells at concentrations which had no significant influence on their vitality (Figure 8). This effect was observed in a dose-dependent fashion in both unactivated and OKT-3-activated T cells. Overall, the cell sizes of the activated T cells decreased more than those of the unactivated T cells; however, the activated T cells were generally larger than the unactivated T cells. The decrease in cell sizes was already statistically significant at concentrations of 0.625 µM and 2.5 µM indisulam (Figure 8a).

Similar effects of indisulam on cell sizes were seen in gp100-activated T cells and moDCs.

In conclusion, analyzing the impact of indisulam on the non-specific and gp100-specific activation of CD4^+^ or CD8^+^ T cells, we detected dose-dependent and time-dependent effects on the upregulation of activation markers and the secretion of cytokines. Indisulam treatment showed a stronger influence on the weaker non-specific T cell activation than on the stronger gp100-specific activation. CD8^+^ T cells seemed to be more sensitive to indisulam than CD4^+^ T cells. Substantial effects on the vitality of T cells were only detected at a concentration of 160 µM indisulam. Interestingly, it was noticeable that T cells shrank in the presence of increasing indisulam concentrations.

### 3.4. Influence of Indisulam on moDC Maturation

In addition to the impact of indisulam on the CD4^+^ and CD8^+^ T cell activation, we analyzed whether indisulam interfered with the maturation of DCs. DCs play a central role in the presentation of antigens and the activation of T cells. They are also the most commonly used antigen-presenting cells in therapeutic cancer vaccination, a form of immunotherapy that trains the patient’s immune system to specifically recognize and attack cancer cells [19]. In our experiments, we used ex vivo-generated moDCs from voluntary donors as model.

#### The Upregulation of Maturation Markers on moDCs Is Inhibited by High Indisulam Concentrations

Initially, we explored whether indisulam treatment affected the maturation of DCs. Experiments were performed with moDCs generated from monocytes of healthy donors and a one-week differentiation protocol with the addition of IL-4 and GM-CSF. On day 5 of the differentiation protocol, immature DCs were treated with DMSO solvent control or with indisulam at the following different concentrations: 0.625 µM, 2.5 µM, 10 µM, or 40 µM. Since we observed heavy adverse effects at an indisulam dose of 160 µM in all our T cell experiments, we decided to go up to 40 µM only for the subsequent experiments. Following 24 h of treatment with indisulam or DMSO, on day 6, the maturation process was induced by a cytokine cocktail containing IL-1β, PGE_2_, IL-6, and TNFα [20]. After 24 h of maturation, the influence of indisulam was tested in assays meant to detect changes in the maturation markers. The moDCs were harvested, stained with a live/dead marker, and assessed for the surface expression of the maturation markers CD25, CD83, CD86, and PD-L1 by flow cytometry.

The data showed a dose-dependent decrease in CD25, CD83, CD86, and PD-L1 expression with increasing concentrations on moDCs treated with indisulam before and during maturation. Overall, the donor variance was relatively high. But considering the mean values in the maturation marker profile, the following observations were made: both PD-L1 and CD25 expression decreased statistically significantly at 10 µM indisulam and more. A slight but significant inhibition on the upregulation of CD83 and CD86 was observed at 0.625 µM, but not all higher indisulam doses showed a formal statistical significance (Figure 9, Appendix A). At 40 µM, the expression of CD83 clearly dropped further, but not that of CD86. These results indicate that concentrations below 10 µM seem to be tolerated rather well, but higher concentrations impair the maturation of the DCs.

### 3.5. Influence of Indisulam on Antigen-Specific CD8^+^ T Cell Priming

To enable T cells to recognize the induced neoantigens on the tumor surface and at-tack cancer cells, T cells need to be primed first. This requires an ensemble of antigen-presenting cells such as DCs and naïve T cells. Therefore, we investigated the impact of indisulam on moDC-mediated priming of CD8^+^ T cells. The antigen-specificity of CD8^+^ T cells was determined by MHC tetramer staining. As described above, 24 h before and during maturation by cytokine cocktail, moDCs were pretreated with DMSO solvent controls or pretreated with indisulam at the following different concentrations: 0.625 µM, 2.5 µM, 10 µM, or 40 µM. After 24 h of maturation, moDCs were harvested and either loaded with a peptide from the tumor antigen MelanA or left unloaded as controls. For the priming approach, over the course of one week, loaded and unloaded moDCs were co-incubated with autologous lymphocytes containing CD8^+^ T cells, in the absence or continuous presence of indisulam at the indicated concentrations. On day 7 of priming, the cells were harvested and stained for CD8, CD4, CCR7, CD45RA, and a live/dead marker. Additionally, the cells were stained with a fluorescently labeled MelanA/MHC tetramer or dextramer to determine the percentage of MelanA-specific CD8^+^ T cells by flow cytometry (Figure 10).

Our data indicate that the priming of CD8^+^ T cells was still possible at indisulam doses of 0.625 µM and 2.5 µM, but was almost completely suppressed at 10 µM and above (Figure 10a,b). These results indicate that T cell priming seems to be more sensitive to indisulam than the mere activation of T cell-effector functions and the maturation of moDCs.

Furthermore, we evaluated the differentiation of CD8^+^ T cells under the influence of indisulam. In general, the differentiation of naïve T cells into memory cells and effector cells can be determined by the changing pattern of CCR7, a chemokine receptor which enables migration to lymph nodes, and CD45RA, the long isoform of a cell surface tyrosine phosphatase. Naïve T cells are characterized by CCR7^+^ CD45RA^+^. Upon antigen stimulation, they lose CD45RA and differentiate into central memory cells, defined as CCR7^+^ CD45RA^−^. Then, following further stimulation, these cells differentiate into effector memory cells, described as CCR7^−^ CD45RA^−^. Ultimately, the final stage of CD8^+^ T cell differentiation results in so-called lytic effector cells, marked as CCR7^−^ CD45RA^+^ [21,22]. Our evaluations showed a shift in differentiation of the specific CD8^+^ T cell following indisulam treatment, even at concentrations that did not influence the overall number of specific T cells. Treatment with 0.6 µM and 2.5 µM indisulam evoked a larger central memory cell population, compared to the DMSO control and lower numbers of effector memory and lytic effector cells. Higher concentrations of indisulam, that blocked expansion, also blocked the differentiation of the T cells, indicated by a large proportion of T cells with the naïve phenotype under the influence of 10 µM and 40 µM indisulam (Figure 10c).

Taken together, indisulam also had a clear effect on T cell priming and differentiation, which could have implications for its use combined with immunotherapeutic strategies.

## 4. Discussion

In the past, immunotherapies have made excellent progress in cancer treatment and have demonstrated that the body’s own immune system can actually be forced to fight against tumors. Ongoing research and the development of new therapeutic approaches aim to unlock the full potential of immunotherapy to improve treatment outcomes for tumors lacking treatment options. The success of immunotherapies correlates with the neoantigen load of tumor cells, which can result from aberrant RNA splicing. In cancer cells, neoantigens primarily originate from somatic DNA mutations, but can also be caused by splicing inhibitors, e.g., indisulam, for therapeutic purposes [4]. Therefore, pharmacologically induced modulation of RNA splicing may offer an untapped source to increase the presentation of immunogenic neoantigens on the tumor surface and broaden the effectiveness of immunotherapies. When considering the idea of administering indisulam to cancer patients to generate immunogenic neoantigens on tumor cells and combining indisulam with immunotherapeutic approaches, it must be ensured that the applied dose of indisulam does not diminish immunocompetence and the power of immunotherapies. To our knowledge, this is the first systematic study on the influence of indisulam on human immune effector cells.

Indisulam has already been clinically evaluated as a cytostatic agent in several phase I and II studies, including patients with refractory solid tumors. By interfering with fundamental cellular functions, in particular cell cycle regulation and RNA splicing, it has been found that indisulam can cause an irreparable loss of function leading to cell death (apoptosis). For example, indisulam may block the G1/S-transition by inhibiting the activation of important cell cycle regulators like cyclin-dependent kinase 2 and cyclin E, reduce the phosphorylation of retinoblastoma protein (pRb), and be associated with the upregulation of p53 [23]. With an intended use as a cytostatic agent, indisulam was administered at high doses. The recommended dose for indisulam as a monotherapy, which was considered safe, was 700 mg/m^2^, resulting in peak serum levels of approximately 130 to 300 µM. These descended rapidly to 90 µM on average after 12 h and gradually decreased further over the next days. The dose-limiting adverse events were reversible neutropenia, thrombocytopenia, and anemia. Most other observed toxicities were rated as mild to moderate, including nausea, fatigue, acne-like rashes, mucositis, conjunctivitis, and alopecia [23]. Even though indisulam was found to be well tolerated in general, there was a consensus that it had only minor single-agent activity [24,25,26]. This encourages the endeavors to combine indisulam with other therapeutic approaches.

With the intention of using indisulam as a splicing modulator for the induction of neoantigens, even low concentrations of indisulam are sufficient. In mice, Lu et al. showed that administering 25 mg/kg indisulam led to neoantigen generation with subsequent CD8^+^ T cell-mediated tumor growth suppression. As expected, with low levels of indisulam, side effects were minimal [10]. To learn more about the potential feasibility of combining immunotherapy and indisulam in a clinical setting, it is crucial to investigate the effect of indisulam on human immune effector cells, since these are important for ensuring maintained immunocompetence, and may react differently than mouse immune cells.

Addressing the general question of whether indisulam treatment impairs the immunocompetence of humans, we examined ex vivo-isolated CD4^+^ and CD8^+^ T cells and moDCs from healthy blood donors to explore the dose-dependent effects of indisulam on T cell activation, moDC maturation, and antigen-specific T cell priming. To cover a broad spectrum, the indisulam concentrations tested in our study ranged from low doses up to very high doses. The maximum doses were based on the observed plasma levels in patients treated with indisulam monotherapy [23]. The selection of our concentrations allowed us to demonstrate dose-dependent effects of indisulam and gave insights into the potential feasibility of future combinations with immunotherapies. Overall, our results showed that indisulam influences different immune cells and different processes within the immune system to varying extents. This provides information on aspects that must be considered for possible future combinations with immunotherapies.

### 4.1. Effects on T Cell Activation and Function

Lu et al. explored the influence of indisulam on murine splenic naïve CD8^+^ T cells and demonstrated that doses of 10 µM had only minimal effects on the non-specific CD8^+^ T cell activation [10]. These findings were in line with our results on ex vivo-isolated CD8^+^ T cells from human blood donors and were additionally shown for CD4^+^ T cells. In general, we observed that CD4^+^ T cells were more robust to indisulam application. In further assays, we showed that the antigen-specific activation of T cells was less affected by indisulam treatment. This raises optimism that indisulam does not interfere with T cells in a manner that prevents T cells from being activated by indisulam-induced neoantigens. To obtain an impression of relevance of the duration of exposure, we tested the influence of indisulam on the non-specific and antigen-specific activation of the more sensitive CD8^+^ T cells for two time points: after 24 h and after 48 h of pretreatment with indisulam. Our evaluations indicated that the observed effects after 48 h of indisulam treatment were in line with the results for after 24 h of indisulam treatment, but more pronounced. In particular, cytokine secretion was impaired and drastically decreased after 48 h of indisulam treatment. In particular, production of IFNγ, a cytokine of high relevance in tumor immunology and in cytotoxicity, was strongly inhibited.

Addressing the crucial question whether CD8^+^ T cells are still capable of killing cancer cells after treatment with indisulam, we determined the extent of tumor lysis. Our data showed that tumor killing was possible with concentrations of up to 40 µM indisulam and was diminished at 160 µM indisulam. These findings suggest that the observed changes in activation markers and cytokine secretion did not translate into functional deficits of T cells.

Remarkably, our analyses showed that indisulam led to cell shrinkage at concentrations which had no significant influence on their vitality. So far, we cannot provide a clear mechanistic explanation for why this occurs. But, there are several published studies with other splicing inhibitors reporting that splicing of pre-mRNA and transcription are closely coordinated [27,28]. Assuming that indisulam may act similarly, a potential approach to explaining the observed cell shrinkage may be that indisulam interferes with RNA transcription in a similar fashion and downregulates protein production in a manner that leads to cell size reduction, but not cell death. However, this remains speculative and requires further investigation.

### 4.2. Effects on DC Maturation and T Cell Priming

For the evaluation of DCs, we performed experiments on moDCs. These were generated following a well-established procedure and served as a model, which is appropriate for ex vivo-analyses of human DCs. However, given the fact that a model was used, it must be noted that our observations may not be directly transferable to other types of DCs in vivo. Our evaluation focused on indisulam’s influence on cytokine-mediated moDC maturation. These indicate that moDCs are more sensitive to indisulam than T cells. The upregulation of the maturation markers CD83 and CD86 was already slightly impaired at an administered dose of 0.625 µM, while the expression of CD25 and PD-L1 decreased at 10 µM indisulam. Further, our investigations showed that indisulam particularly impedes the priming of T cells by peptide-loaded moDCs.

### 4.3. Implications for Combinations with Immunotherapy

In summary, our data suggest that combining indisulam treatment with immunotherapeutic approaches is potentially feasible. However, we identified dose-dependent effects of indisulam on different immune cells. To avoid compromising the therapeutic success of immunotherapy, the choice of immunotherapy and therapeutic setting should be reconciled with the indisulam dose that is required for neoantigen induction and with which immune processes might be affected in the presence of this indisulam level.

Since treatment with indisulam barely affects antigen-specific T cell activation, a combination with adoptive cell therapy using TCR-engineered T cells that target neoantigens evoked by indisulam-induced splicing modulation is expected to be a feasible option. Another possibility is the combination of indisulam with DC vaccines. The production of DC vaccines implies antigen loading and DC maturation ex vivo. In the case of pairing a DC vaccine with indisulam treatment, an indisulam-induced neoantigen is utilized. When indisulam and the DC vaccine are co-administered, the indisulam concentration must not hinder the injected antigen-loaded DCs from priming CD8^+^ T cells in vivo and evoking an anti-tumor response. As we have seen, DCs, especially with regard to T cell priming, are sensitive to indisulam; this therapeutic setting would thus only tolerate low doses of indisulam. However, to circumvent this issue, it is also possible to perform the treatment with the DC vaccine prior to indisulam administration. In this case, CD8^+^ T cells are primed by antigen-loaded DCs still in the absence of the inhibitor. Later, higher indisulam concentrations can be administered to the patient to induce in vivo aberrant splicing and subsequent generation of the neoantigen that had been used for the DC vaccine. Another possibility would involve performing T cell priming ex vivo. This therapeutic setting would also tolerate higher indisulam concentrations, because T cells are primed first in the absence of indisulam and are then injected into the patient after indisulam-induced neoantigen formation.

## 5. Conclusions

Taken together, this study underlines the potential feasibility of combining indisulam with other immunotherapeutic approaches. We provide insights into the dose-dependent influence of indisulam on different immune processes. Our evaluation highlights the need to align the choice of immunotherapy with the indisulam dose required for the induction of splicing-derived neoantigens.

## Figures and Tables

**Figure 1 pharmaceutics-17-00368-f001:**
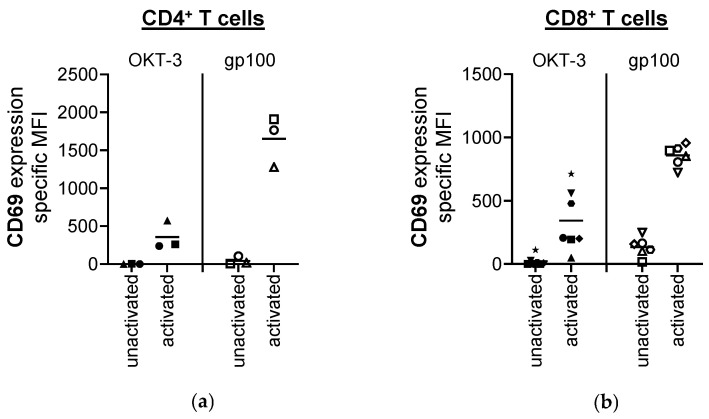
Antigen-specific activation is generally stronger than the non-specific activation of T cells. Displayed are CD4^+^ T cells (**a**) and CD8^+^ T cells (**b**), which were left untreated (no indisulam, no DMSO). The non-specific T cell activation was performed by CD3-crosslinking (OKT-3). Antigen-specific T cell activation was performed with T cells transfected with a gp100-specific TCR by co-incubating them with gp100-loaded target cells (gp100). After 20–24 h of activation, T cells were harvested, and living T cells were analyzed for the surface expression of activation markers using flow cytometry. Here, the expression of CD69 is shown as an example. The specific MFIs were calculated by subtracting the background MFI of the unstained controls and mean values (horizontal bars) are shown from three, six, or seven different donors (represented as different symbols). Similar observations were made for the expression of CD25.

**Figure 2 pharmaceutics-17-00368-f002:**
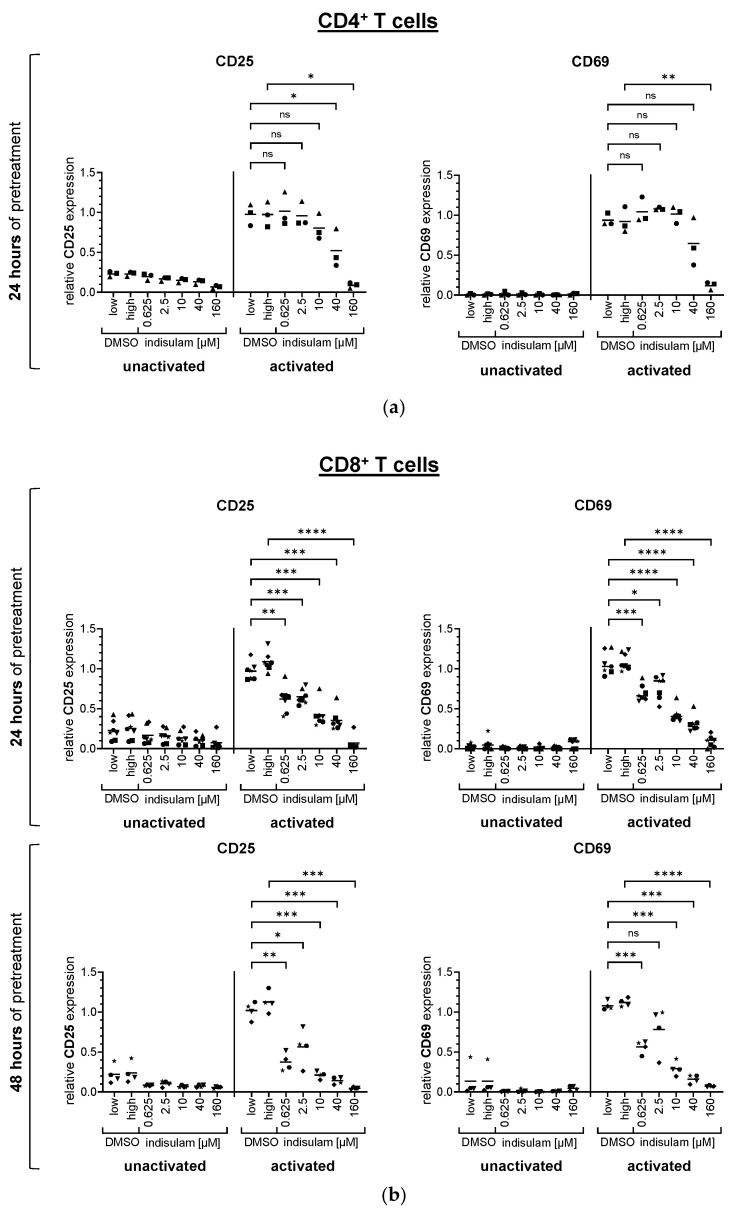
Indisulam influences the expression of the activation markers CD25 and CD69 on T cells following activation by CD3-crosslinking. CD4^+^ (**a**) and CD8^+^ (**b**) T cells were either treated with the splicing inhibitor indisulam at the indicated concentrations, treated with the DMSO solvent controls (DMSO low equivalent to the quantity of DMSO contained in 40 µM indisulam, DMSO high equivalent to the quantity of DMSO contained in 160 µM indisulam), or left untreated. After treatment of approximately 24 h or 48 h, T cells were activated in the presence or absence of DMSO or indisulam by CD3-crosslinking with the agonistic antibody OKT3 or were left unactivated. After 20–24 h of activation, T cells were harvested, and living T cells were analyzed for surface expression of the activation markers CD25 and CD69 using flow cytometry. Specific MFIs were calculated by subtracting the background MFI of the respective unstained sample. The figure depicts the relative expression of the surface markers CD25 and CD69, calculated by relating the specific MFI to the reference value of T cells activated in absence of DMSO or indisulam. Mean values (horizontal bars) are shown from three (**a**), four ((**b**), lower panel), or seven ((**b**), upper panel) different donors (represented as different symbols). *p*-values were calculated with paired Student’s *t*-tests using the values for relative expression compared to the respective DMSO solvent control. The conditions up to 40 µM indisulam were tested against DMSO low, and the condition with 160 µM indisulam was tested against DMSO high. * = significant (* *p* ≤ 0.05, ** *p* ≤ 0.01, *** *p* ≤ 0.001, **** *p* ≤ 0.0001); ns = not significant (*p* > 0.05).

**Figure 3 pharmaceutics-17-00368-f003:**
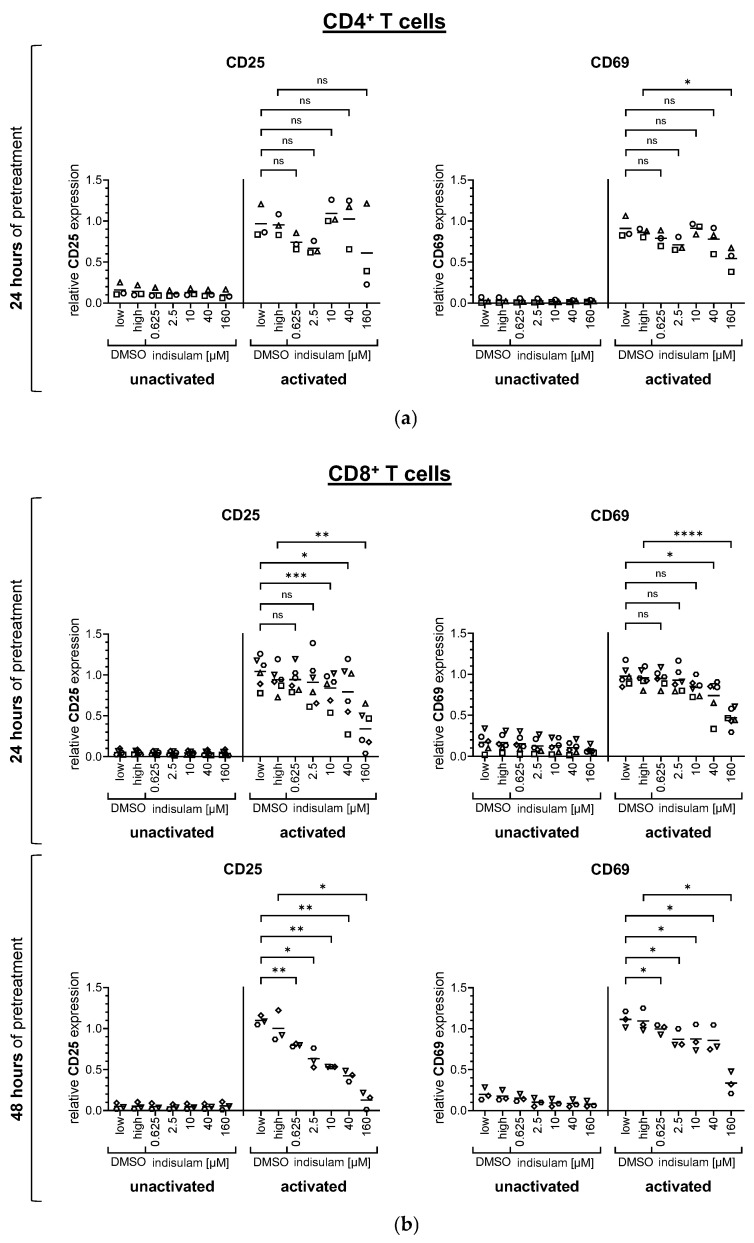
Indisulam has a weaker influence on the expression of the activation markers CD25 and CD69 on T cells following antigen-specific activation. CD4^+^ (**a**) and CD8^+^ (**b**) T cells were equipped with a gp100-specific TCR by mRNA electroporation and then either treated with the splicing inhibitor indisulam at the indicated concentrations, treated with the DMSO solvent controls (DMSO low equivalent to the quantity of DMSO contained in 40 µM indisulam, DMSO high equivalent to the quantity of DMSO contained in 160 µM indisulam), or left untreated. After treatment of approximately 24 h or 48 h, T cells were co-incubated in the presence or absence of DMSO or indisulam with T2.A1 target cells, that had been loaded with the gp100 peptide (activated) or had been left unloaded as controls (unactivated). After 20–24 h of activation, T cells were harvested and living T cells were analyzed for the surface expression of the activation markers CD25 and CD69 using flow cytometry. Specific MFIs were calculated by subtracting background MFI of the respective unstained sample. The figure depicts the relative expression of the surface markers CD25 and CD69, calculated by relating the specific MFI to the reference value of T cells activated in absence of DMSO or indisulam. Mean values (horizontal bars) are shown from three, as in (**a**) and ((**b**), lower panel), or six ((**b**), upper panel) different donors (represented as different symbols). *p*-values were calculated with paired Student’s *t*-tests using the values for relative expression compared to the respective DMSO solvent control. The conditions up to 40 µM indisulam were tested against DMSO low, and the condition with 160 µM indisulam was tested against DMSO high. * = significant (* *p* ≤ 0.05, ** *p* ≤ 0.01, *** *p* ≤ 0.001, **** *p* ≤ 0.0001); ns = not significant (*p* > 0.05).

**Figure 4 pharmaceutics-17-00368-f004:**
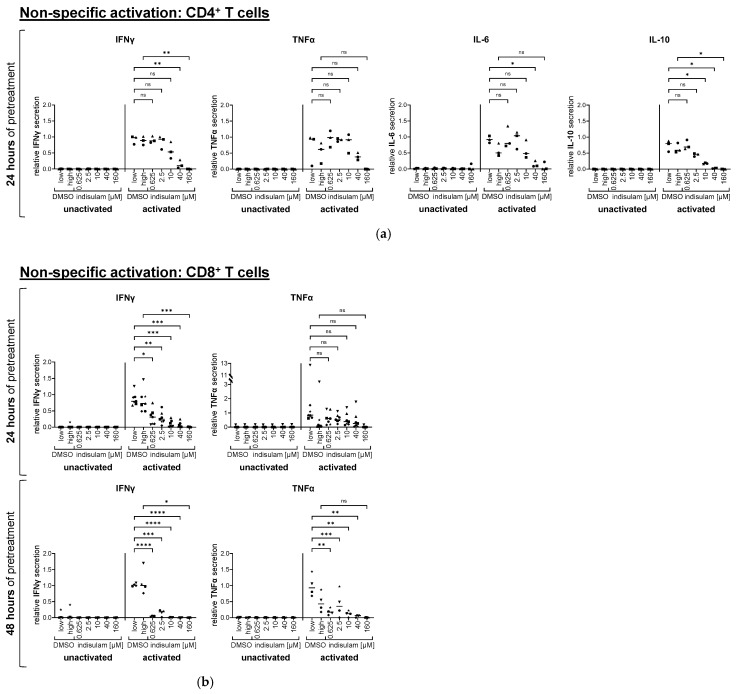
Indisulam affects cytokine secretion by T cells activated by CD3-crosslinking in a dose- and time-dependent manner. CD4^+^ (**a**) and CD8^+^ (**b**) T cells were either treated with the splicing inhibitor indisulam at the indicated concentrations, treated with the DMSO solvent controls (with DMSO low being equivalent to the quantity of DMSO contained in 40 µM indisulam, and DMSO high being equivalent to the quantity of DMSO contained in 160 µM indisulam) or were left untreated (i.e., no indisulam, no DMSO). After treatment for approximately 24 h or 48 h, T cells were activated by CD3-crosslinking in the presence or absence of DMSO or indisulam for 20–24 h. Supernatants were collected and analyzed for secreted cytokines. The figure shows the relative cytokine concentration, calculated by relating the respective cytokine concentration to the reference value of T cells activated in absence of DMSO or indisulam. Mean values (horizontal bars) are shown from three (**a**), seven ((**b**), upper panel), or four ((**b**), lower panel) different donors (represented as different symbols). *p*-values were calculated with paired Student’s *t*-tests using the values for relative concentrations compared to the respective DMSO solvent control. Conditions of up to 40 µM indisulam were tested against DMSO low, and the condition with 160 µM indisulam was tested against DMSO high. * = significant (* *p* ≤ 0.05, ** *p* ≤ 0.01, *** *p* ≤ 0.001, **** *p* ≤ 0.0001); ns = not significant (*p* > 0.05).

**Figure 5 pharmaceutics-17-00368-f005:**
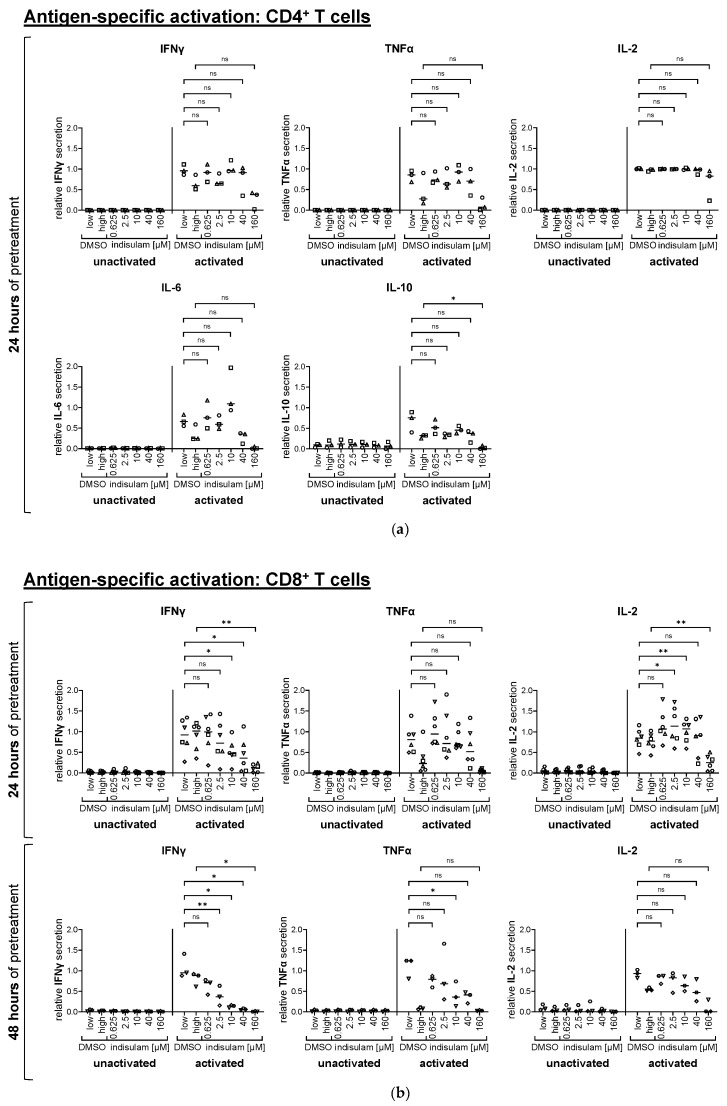
Indisulam has a weaker influence on the cytokine secretion by T cells following antigen-specific activation. CD4^+^ (**a**) and CD8^+^ (**b**) T cells were equipped with a gp100-specific TCR by mRNA electroporation and then either treated with the splicing inhibitor indisulam at the indicated concentrations, treated with the DMSO solvent controls (DMSO low equivalent to the quantity of DMSO contained in 40 µM indisulam, DMSO high equivalent to the quantity of DMSO contained in 160 µM indisulam), or left untreated. After treatment of approximately 24 h or 48 h, T cells were co-incubated in the presence or absence of DMSO or indisulam with T2.A1 target cells that had been loaded with the gp100 peptide (activated) or had been left unloaded as control (unactivated). After 20–24 h of activation, supernatants were collected and analyzed for secreted cytokines. The figure shows the relative cytokine concentration, calculated by relating the respective cytokine concentration to the reference value of T cells activated in absence of DMSO or indisulam. Mean values (horizontal bars) are shown from three (**a**), six ((**b**), upper panel), or three ((**b**), lower panel) different donors (represented as different symbols). *p*-values were calculated with paired Student’s *t*-tests using the values for relative concentration compared to the respective DMSO solvent control. Conditions of up to 40 µM indisulam were tested against DMSO low, and the condition with 160 µM indisulam was tested against DMSO high. * = significant (* *p* ≤ 0.05, ** *p* ≤ 0.01); ns = not significant (*p* > 0.05).

**Figure 6 pharmaceutics-17-00368-f006:**
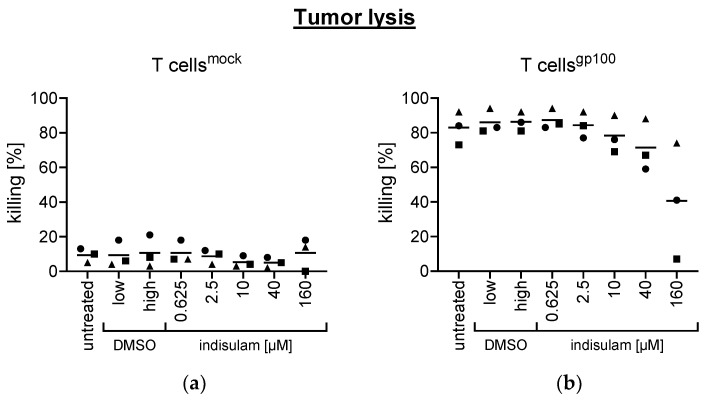
Antigen-specific tumor lysis by indisulam-pretreated T cells remains robust, but is suppressed at the highest indisulam concentration. gp100-TCR-electroporated CD8^+^ T cells (T cells^gp100^) and mock-electroporated CD8^+^ T cells (T cells^mock^) were either treated with the splicing inhibitor indisulam at the indicated concentrations, treated with the DMSO solvent controls (DMSO low equivalent to the quantity of DMSO contained in 40 µM indisulam, and DMSO high equivalent to the quantity of DMSO contained in 160 µM indisulam), or were left untreated (i.e., no indisulam, no DMSO). After treatment of approximately 24 h, T cells were co-cultured overnight with gp100-positive target tumor cells and gp100-negative control tumor cells. The next day, cells were harvested, stained with the Zombie NIR-live/dead marker and analyzed for tumor killing. Data analysis was performed by determining the percentage of living target tumor cells and living control tumor cells. Living tumor cells (i.e., Zombie NIR-negative tumor cells) were determined by flow cytometry. The tumor killing percentage was calculated by using the lysis formula as indicated in the materials and methods section. (**a**) The tumor killing percentage by mock-electroporated CD8^+^ T cells (T cells^mock^) after no pretreatment or pretreatment with DMSO or indisulam at different concentrations of a wide range (as indicated). (**b**) Tumor killing percentage by gp100-TCR-electroporated CD8^+^ T cells (T cells^gp100^) after no pretreatment or treatment with DMSO or indisulam at a wide range of different concentrations (as indicated). The tumor killing percentages and mean values (horizontal bars) are shown from three different donors (represented as different symbols). *p*-values were calculated with paired Student’s *t*-tests using the values for tumor killing percentage compared to the respective DMSO solvent control. The conditions up to 40 µM indisulam were tested compared to DMSO low, and the condition with 160 µM indisulam was tested compared to DMSO high.

**Figure 7 pharmaceutics-17-00368-f007:**
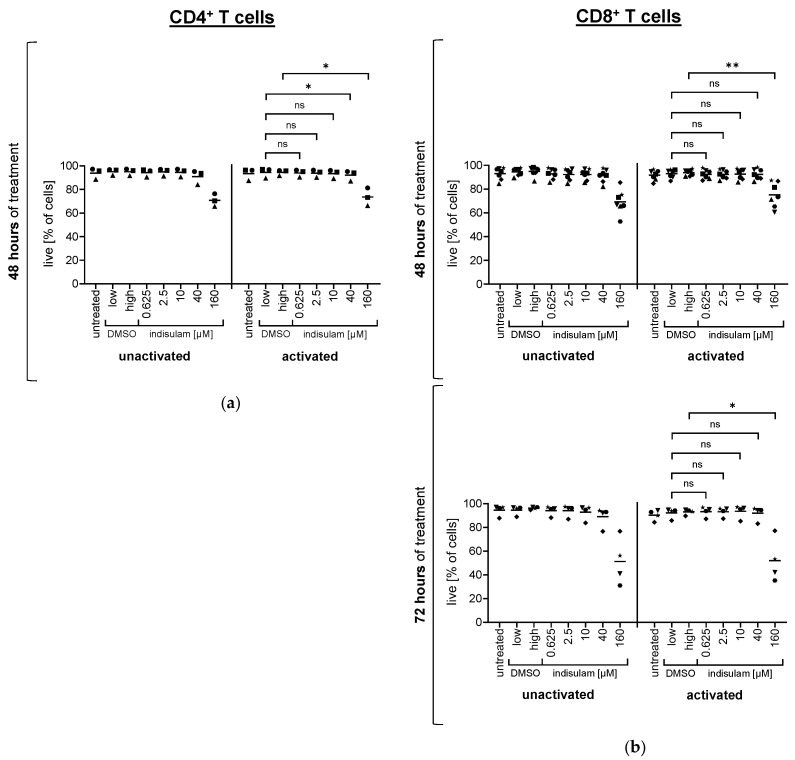
Indisulam does not reduce the vitality of T cells up to a dose of 40 µM, but toxic effects are observed at an indisulam concentration of 160 µM. CD4^+^ (**a**) and CD8^+^ (**b**) T cells were either treated with the splicing inhibitor indisulam at the indicated concentrations, treated with the DMSO solvent controls (DMSO low being equivalent to the quantity of DMSO contained in 40 µM indisulam, and DMSO high being equivalent to the quantity of DMSO contained in 160 µM indisulam), or were left untreated (i.e., no indisulam, no DMSO). After treatment of approximately 24 h or 48 h, T cells were either left unactivated or were activated by CD3-crosslinking in the presence or absence of DMSO or indisulam. After another 20–24 h, T cells were harvested, stained with the 7-AAD-live/dead marker, and analyzed for their vitality. Living cells (i.e., 7-AAD-negative cells) were determined by flow cytometry. The percentage of living cells was obtained based on the number of 7-AAD-negative cells among all cells. Mean values (horizontal bars) are shown from three (**a**), seven ((**b**), upper panel), or four ((**b**), lower panel) different donors (represented as different symbols). *p*-values were calculated with paired Student’s *t*-tests using the values for the percentage of living cells compared to the respective DMSO solvent control. The conditions up to 40 µM indisulam were tested compared to DMSO low, and the condition with 160 µM indisulam was tested compared to DMSO high. * = significant (* *p* ≤ 0.05, ** *p* ≤ 0.01); ns = not significant (*p* > 0.05).

**Figure 8 pharmaceutics-17-00368-f008:**
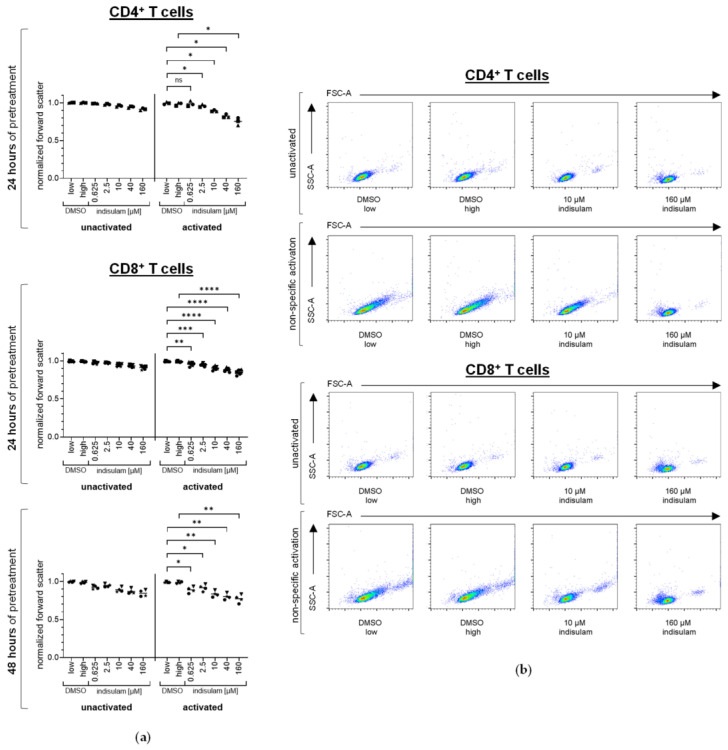
T cells shrink in the presence of increasing indisulam concentrations. CD4^+^ and CD8^+^ T cells were either treated with the splicing inhibitor indisulam at the indicated concentrations, treated with the DMSO solvent controls (with DMSO low being equivalent to the quantity of DMSO contained in 40 µM indisulam, and DMSO high being equivalent to the quantity of DMSO contained in 160 µM indisulam) or were left untreated (i.e., no indisulam, no DMSO). After treatment of approximately 24 h or 48 h, T cells were either left unactivated or were activated by CD3-crosslinking in the presence or absence of DMSO or indisulam. After another 20–24 h, T cells were harvested, stained with the 7-AAD-live/dead marker and analyzed for their cell size by flow cytometry (using a sideward scatter area (SSC-A) and a forward scatter area (FSC-A)). (**a**) Forward scatter of living, unactivated, or non-specifically activated CD4^+^ or CD8^+^ T cells after treatment with DMSO or indisulam at different concentrations. The evaluation of cell sizes was performed by normalizing the MFI of FSC-A to the reference value. Mean values (horizontal bars) are shown from three ((**a**), upper panel), seven ((**a**), middle panel), or four ((**a**), lower panel) different donors (represented as different symbols). *p*-values were calculated with paired Student’s *t*-tests, using the values for normalized FSC-A compared to the respective DMSO solvent control. The conditions up to 40 µM indisulam were tested compared to DMSO low, and the condition with 160 µM indisulam was tested compared to DMSO high. * = significant (* *p* ≤ 0.05, ** *p* ≤ 0.01, *** *p* ≤ 0.001, **** *p* ≤ 0.0001); ns = not significant (*p* > 0.05). (**b**) Exemplary illustration of dot plots from one donor.

**Figure 9 pharmaceutics-17-00368-f009:**
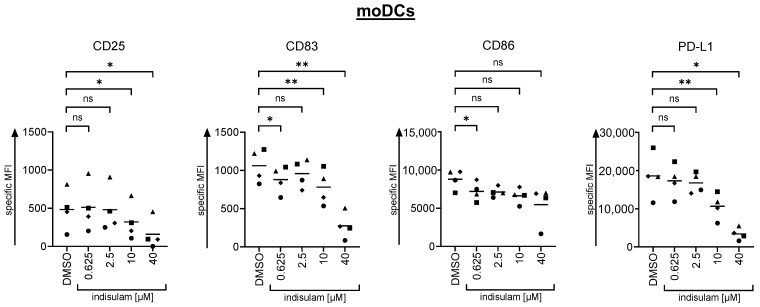
The maturation-mediated upregulation of certain surface markers on moDCs is inhibited by high indisulam concentrations, but only moderately by low indisulam concentrations. Immature moDCs were treated with the splicing inhibitor indisulam at the indicated concentrations or treated with the DMSO solvent control (equivalent to the quantity of DMSO contained in the condition with 40 µM indisulam). After 24 h of treatment, moDCs were matured in the presence or absence of indisulam with a standard cytokine cocktail containing IL-1β, PGE_2_, IL-6, and TNFα. After 24 h of maturation, moDCs were harvested and analyzed for surface expression of the maturation markers CD25, CD83, CD86, and PD-L1 using flow cytometry. Specific MFIs were calculated by subtracting the background MFI of the unstained controls and mean values (horizontal bars) are shown from four experiments (represented as different symbols). *p*-values were calculated with paired Student’s *t*-tests using the values for a specific MFI compared to the specific MFI of the DMSO solvent control. * = significant (* *p* ≤ 0.05, ** *p* ≤ 0.01); ns = not significant (*p* > 0.05).

**Figure 10 pharmaceutics-17-00368-f010:**
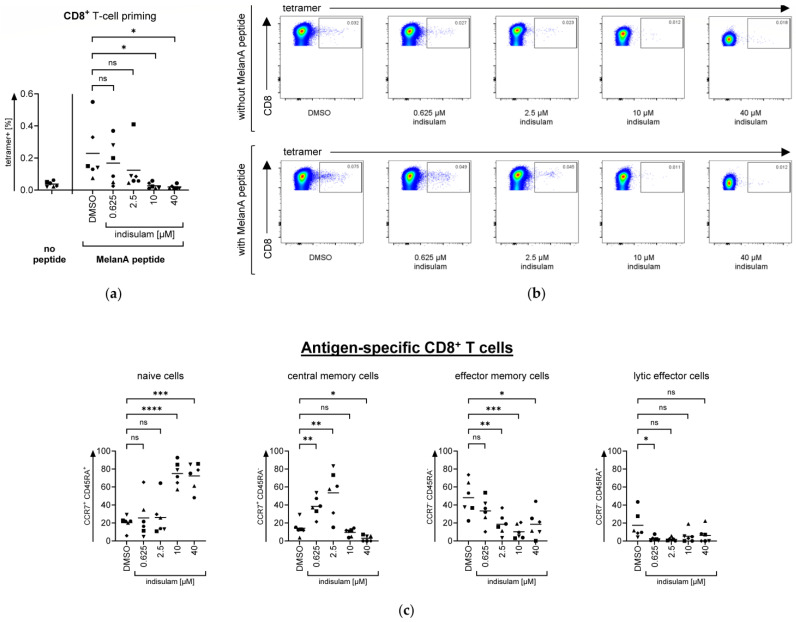
moDC-mediated priming of CD8^+^ T cells is affected at low indisulam concentrations and suppressed at higher concentrations. Immature moDCs were pretreated with the splicing inhibitor indisulam at the indicated concentrations or pretreated with the DMSO solvent control (equivalent to the quantity of DMSO contained in 40 µM indisulam). After 24 h of pretreatment, moDCs were matured in the presence or absence of indisulam with a standard cytokine cocktail containing IL-1β, PGE2, IL-6, and TNFα. After 24 h of maturation, moDCs were either loaded with a peptide from the tumor antigen MelanA or left unloaded as controls. In a one-week priming approach, moDCs were co-incubated with autologous NAF containing CD8^+^ T cells, in the presence or absence of DMSO or indisulam at the indicated concentrations. After one week of incubation, the cells were harvested, stained for CD8, CD4, CCR7, and CD45RA, and analyzed for antigen specificity of CD8^+^ T cells by MHC multimer staining. (**a**) Percentage of MelanA-specific (i.e., tetramer+) CD8^+^ T cells in response to priming by MelanA-loaded moDCs after treatment with DMSO or indisulam at different concentrations of a wide range as indicated. Mean values (horizontal bars) are shown from six different donors (represented as different symbols). The average background (without MelanA peptide) was 0.037%. *p*-values were calculated with paired Student’s *t*-tests using the values for the percentage of MelanA-specific T cells compared to the DMSO solvent control. (**b**) Exemplary illustration of dot plots from one donor. (**c**) Characterization of MelanA-specific naïve, central memory, effector memory, or lytic effector CD8^+^ T cells. Mean values (horizontal bars) are shown from six different donors (represented as different symbols). *p*-values were calculated with paired Student’s *t*-tests using the values for the percentage of cell population compared to the DMSO solvent control. * = significant (* *p* ≤ 0.05, ** *p* ≤ 0.01, *** *p* ≤ 0.001, **** *p* ≤ 0.0001); ns = not significant (*p* > 0.05).

## Data Availability

Data are available upon reasonable request from the corresponding author.

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
