# Peer review of "The Influence of Indisulam on Human Immune Effector Cells: Is a Combination with Immunotherapy Feasible?"

_pharmaceutics, 2025, doi:10.3390/pharmaceutics17030368_

Round 1
Reviewer 1 Report
Comments and Suggestions for Authors
The authors designed this study to evaluate, The influence of the mRNA splicing inhibitor indisulam on human immune effector cells: Is a combination with immune therapy feasible?, the work showed promising outcomes. However, there are a number of minor concerns with the manuscript that need to be addressed.
The authors have focused on exploring the influence of the mRNA splicing inhibitor indisulam on human immune effector cells and its potential combination with immunotherapy. This is a novel and timely topic, particularly with the growing interest in combining targeted therapies with
1. This is a well-designed and well-explained paper, with all necessary details provided in the methodology and introduction.
2. While the study provides interesting observations regarding the effects of indisulam on T cell size, but it does not provide a clear mechanistic explanation for why this occurs. Understanding the underlying biological processes is crucial to interpreting the significance of this finding. For instance, is the shrinkage due to cell cycle arrest, changes in metabolic activity, or alterations in cytoskeletal dynamics? Without mechanistic data, the observed shrinkage might be difficult to place in the context of T cell function and the potential impact of indisulam.
3. It seems 1-hour incubation time for peptide loading might be criticized as too short to allow sufficient peptide-MHC complex formation and surface expression. In some cases, longer incubation periods are needed to ensure adequate presentation. This could be a significant flaw if the resulting peptide presentation is insufficient to elicit a detectable T cell response.
Furthermore, while the authors used unloaded moDCs as negative controls as a standard, but they did not include a positive control with a known strong immunogenic peptide to ensure that the experimental setup and detection methods are functioning correctly. The absence of such controls could leave the experiment vulnerable to false negatives or underperformance of the assay.
4. Why authors did not assess the capacity of T cells to kill cancer cells, following treatment with indisulam? These assays would help determine whether the observed changes in cell size and activation markers translate into functional deficits.
Some suggested methods to make the study more comprehensive and detailed that can be considered for the next study.
I. RNA-Seq: To profile changes in gene expression induced by indisulam in immune effector cells, focusing on splicing alterations.
II. qRT-PCR: To validate specific gene targets identified by RNA-Seq, particularly those involved in immune regulation and function.
III. In Vivo Models: Use mouse models to assess the therapeutic efficacy of the combination treatment, including tumor growth inhibition and survival studies.
IV. Using ex vivo tumor models, such as organoids or patient-derived tumor explants, to assess the effects of indisulam on immune cell infiltration and function within a more clinically relevant setting.

Reviewer 2 Report
Comments and Suggestions for Authors
It is acceptable in its current state.
Reviewer 3 Report
Comments and Suggestions for Authors
Overall, this paper was hard to follow and the conclusions were weak at best given the low sample number and the sample to sample variation. I am not sure the paper adds much to the field and also do not think the results support the conclusion of combining indisulam being with immunotherapy. The variation of response based on different doses (but not being linked to a dose-response curve) makes dosing this drug to response tricky.
Some specific comments: lines 311-332 describe data but no figure reference is given. Lines 328-329 mention cytokine data but again, no figure reference is provided. Lines 331-333, I am not sure I see where this comparison is made on Figure 1. Overall, the data description and figure layout of Figure 1 is very confusing.
To strengthen this paper, more samples are needed. Samples should also be assessed at multiple time points to understand what is happening over time. Furthermore, data beyond just cell surface marker expression and cytokines would also help this study.
Reviewer 4 Report
Comments and Suggestions for Authors
Comments to Authors
The research article by Arnet et al described the role of indisulam as cancer immunotherapy and its effectiveness as a therapeutic approach for the control of tumorigenesis.
The current manuscript shows the necessary aspects of indisulam, a carbonic anhydrase inhibitor and a potential anti-cancer agent in cancer immunotherapy, this topic is essential, nevertheless, below are some critical revisions that need to be addressed before publication is moved forward.
1. Indisulam and its potential anti-cancer are well-known, so particularly this reviewer thinks this is a sort of repetition of the previous studies, and overall novelty is missing.
2. Again, the rationale behind indisulam in modulating effector immune cells is missing, and it's unclear from the introduction or abstract.
3. The title of the manuscript is too long.
4. For Figure 1, the CD25, though is IL-2RA, it is mostly expressed on regulatory T cells, the Ki67 is the better marker for the activation and proliferation. Additionally, the graph plotted with relative expression, since this is flow cytometry, the graph should be presented with MFI or percent positive population.
5. Figure 2 is confusing, author showed IL-10 is more affected, however, TNFa and IFNg are drastically affected too with high concentrations of indisulam. Furthermore, IL-10 is not just the only anti-inflammatory cytokine, TGFb, IL-13, IL-35 are anti-inflammatory too.
6. The relative expression in graphical representation is confusing, this reviewer prefers to present in MFIs or percent-positive cells.
7. For Figure 3, indisulam concentration used, 2.5, 10, 40 um to directly 160 um, what about 60, 80 or 100, 120 um concentrations? And what could be the mechanism that indisulam affects viability at higher concentrations.
8. What could be the possible mechanism for “T cells shrink in the presence of increasing indisulam concentrations”, and here even with lower concentrations of indisulam T cells shrink, but not affects viability as per Figure 3, it’s confusing.
9. Yet again the rationale behind checking the expression of CD25 and PD-L1 on DC’s is confusing, generally CD25 and PD-L1 are exhaustion markers
10. Author showed the priming of CD8 T cells, they should show the effector and cytokine activity of the CD8 T cells in the same setting.
11. Several grammatical and typographical mistakes need to be corrected.
Comments on the Quality of English Language
Several grammatical and typographical mistakes need to be corrected.
Round 2
Reviewer 3 Report
Comments and Suggestions for Authors
No further comments. Great job addressing the reviews and adding in more data.
Reviewer 4 Report
Comments and Suggestions for Authors
Authors address comments satisfactorily, the manuscript can be accepted for publication.